

# Antarctic macroalgal-associated amphipod assemblages exhibit long-term resistance to ocean acidification

Hannah E. Oswalt[1], Julie B. Schram[2], Margaret O. Amsler[1], Charles D. Amsler[1] and James B. McClintock[1]

[1] Department of Biology, University of Alabama - Birmingham, Birmingham, AL, United States of America
[2] Department of Natural Sciences, University of Alaska - Southeast, Juneau, AK, United States of America

## ABSTRACT

The pH of the world's oceans has decreased since the Industrial Revolution due to the oceanic uptake of increased atmospheric $CO_2$ in a process called ocean acidification. Low pH has been linked to negative impacts on the calcification, growth, and survival of calcifying invertebrates. Along the Western Antarctic Peninsula, dominant brown macroalgae often shelter large numbers of diverse invertebrate mesograzers, many of which are calcified. Mesograzer assemblages in this region are often composed of large numbers of amphipods which have key roles in Antarctic macroalgal communities. Understanding the impacts of acidification on amphipods is vital for understanding how these communities will be impacted by climate change. To assess how long-term acidification may influence the survival of different members in these assemblages, mesograzers, particularly amphipods, associated with the brown alga *Desmarestia menziesii* were collected from the immediate vicinity of Palmer Station, Antarctica (S64°46′, W64°03′) in January 2020 and maintained under three different pH treatments simulating ambient conditions (approximately pH 8.1), near-future conditions for 2100 (pH 7.7), and distant future conditions (pH 7.3) for 52 days then enumerated. Total assemblage number and the relative proportion of each species in the assemblage were found to be similar across the pH treatments. These results suggest that amphipod assemblages associated with *D. menziesii* may be resistant to long-term exposure to decreased pH.

## INTRODUCTION

Atmospheric $CO_2$ concentrations have increased by approximately 50% since the Industrial Revolution (*Joos & Spahni, 2008*; *Lan & Keeling, 2024*) from human-derived emissions such as the combustion of fossil fuels, the production of cement, and deforestation (*Malhi, Meir & Brown, 2002*; *Heede, 2014*; *Paraschiv & Paraschiv, 2020*). Approximately a third of atmospheric $CO_2$ enters the ocean (*Sabine et al., 2004*; *Gruber et al., 2019*) where it reacts with seawater and ultimately releases free hydrogen ions, thereby lowering pH in a process called ocean acidification (OA). As a result of this process and increasing $CO_2$ emissions,

Corresponding author
Hannah E. Oswalt, heoswalt@uab.edu

the average ocean surface pH has decreased by 0.1 pH units and is predicted to decrease a further 0.4 pH units over the next eighty years (*IPCC, 2022*).

Decreased ocean pH can have a direct impact on the physiology and behavior of marine organisms, especially those that calcify like amphipods (*Kroeker et al., 2013*; *Kindinger, Toy & Kroeker, 2022*). For example, marine invertebrates can be susceptible to internal acidification of body fluids and the dissolution of shells to compensate for internal acidification (*Doney et al., 2009*). Furthermore, OA exposure can lower calcification (*Anand et al., 2021*; *Ramaekers et al., 2023*), increase mortality (*Park et al., 2020*), hinder growth (*Sheppard Brennand et al., 2010*; *Bhuiyan et al., 2022*), increase tissue damage (*Anand et al., 2021*), decrease fertilization (*Kurihara & Shirayama, 2004*; *Ericson et al., 2010*; *Borges et al., 2018*), and lead to developmental abnormalities in invertebrates (*Ericson et al., 2010*; *Agnalt et al., 2013*).

Crustaceans possess exoskeletons that are commonly composed of calcite with a protective outer cuticle that can provide some dissolution protection from OA (*Chave, 1954*; *Boßelmann et al., 2007*; *Whiteley, 2011*; *Leung, Zhang & Connell, 2022*). While the exoskeleton may be more protected from dissolution, the mineralogy of crustacean exoskeletons can be altered from OA exposure (*Siegel et al., 2022*). For example, the shrimp *Lysmata californica* becomes more susceptible to exoskeletal 'cracking' and more vulnerable to predators after an increase in Ca/Mg ratios in the shrimp's exoskeleton following low pH exposure (*Taylor et al., 2015*). Furthermore, exposure to OA conditions can lengthen the intermolt period for some crustaceans and can lead to spikes in mortality when molting (*Keppel, Scrosati & Courtenay, 2012*; *Schram et al., 2016b*; *Long, Swiney & Foy, 2021*).

Most invertebrates can compensate for increased internal proton concentrations by either consuming more food or diverting energy toward compensatory behaviors. For example, larvae of the sea urchin *Strongylocentrotus purpuratus* diverts over 40% of its total ATP to protein synthesis and ion transport under OA conditions (*Pan, Applebaum & Manahan, 2015*). Krill, scallops, mussels, and hard corals have all been found to increase their feeding rates to meet higher metabolic rates and to mitigate decreases in growth and calcification under decreased pH conditions (*Melzner et al., 2011*; *Saba et al., 2012*; *Towle, Enochs & Langdon, 2015*). However, a meta-analysis by *Brown et al. (2018)* found that a higher food supply could exacerbate negative impacts of OA on invertebrate growth. Regardless, merely consuming more food may not provide enough energy for compensatory behaviors, especially if their food source is also impacted by OA (*Schoenrock et al., 2015*; *Oswalt et al., 2025*). This may result in invertebrates reallocating energy away from growth and reproduction potential to be invested in compensatory behaviors (*Whiteley, 2011*; *Mardones et al., 2022*).

Most OA work has focused on single species making current understanding of single organism responses to OA better known than community or ecosystem level responses (*Doney et al., 2009*; *Doney et al., 2020*). However, the response of the community can buffer the impacts of OA on a particular species (*Hendriks, Duarte & Álvarez, 2010*) if biodiversity is high. The insurance effect describes a situation where a community is protected and stabilized by high biodiversity that ensures that a subsection of the community can maintain its functional roles even if another subsection of the community fails (*Yachi &*

*Loreau, 1999*). In practice, high biodiversity has been shown to lower the negative impacts of OA on vulnerable organisms in hard bottom communities by 50–90% depending on the species (*Rastelli et al., 2020*). However, the insurance effect can only be effective if biodiversity remains high. Variation in responses to pH changes will likely result in ecosystems undergoing successive changes of organisms, resulting in an eventual reduction in structural complexity and decreased biodiversity in benthic communities (*Andersson, Mackenzie & Gattuso, 2011*; *Doney et al., 2020*). This reduction in biodiversity can reduce the relative efficiency of resistant species and weaken the insurance effect of biodiversity (*Eklöf et al., 2012*).

Marine environments in high latitudes, such as the waters around the Western Antarctic Peninsula (WAP), are notably vulnerable to decreases in pH from $CO_2$ absorption, which can vary depending on wind patterns (*Conrad & Lovenduski, 2015*). Cold, high latitude waters have a higher Revelle factor (*Revelle & Suess, 1957*; *Sabine et al., 2004*) and, therefore, lower buffering capacity (*Jiang et al., 2019*). Furthermore, saturation states of aragonite and calcite tend to be lower in polar regions due to a higher $K_{sp}$ from lower water temperatures (*Jiang et al., 2015*; *Cai et al., 2021*). Undersaturation events of aragonite already occur in parts of the Southern Ocean (*Hauri, Friedrich & Timmermann, 2016*). Under OA conditions, the extent and duration of these events are projected to increase, resulting in some regions experiencing months of aragonite undersaturation yearly (*Hauri, Friedrich & Timmermann, 2016*). Antarctic invertebrates have been found to be very sensitive to changes in pH, with many species showing negative responses to any $CO_2$ concentrations above present day levels (*Hancock et al., 2020*). However, there is evidence that some Antarctic invertebrates are more resilient to OA. Antarctic crustaceans are considered to be less susceptible to OA compared to more sessile calcifiers, like mollusks (*Figuerola et al., 2021*). However, even sessile organisms can adapt to OA exposure with long-term ecological records showing that the Antarctic clam *Laternula elliptica* can develop its outer organic layer to protect its shell from acidification (*Seo et al., 2024*).

Macroalgae often dominate shallow, hard bottom communities along the WAP (*Wiencke, Amsler & Clayton, 2014*). These macroalgal forests shelter large numbers of mesograzers (*Huang et al., 2007*; *Aumack et al., 2011b*; *Amsler et al., 2022*) in assemblages that mostly consist of amphipods with large numbers of gastropods and smaller numbers of isopods, copepods, and ostracods (*Iken, 1999*; *Huang et al., 2007*; *Schram et al., 2016a*; *Amsler et al., 2022*). The density of amphipods on macroalgae in this region can be extremely high. Stands of *Desmarestia menziesii*, a dominant brown alga, can shelter an estimated 300,000 individuals m$^{-2}$ of the benthic substrate and an estimated 26,000 individuals m$^{-2}$ on the red alga *Plocamium* sp. (*Huang et al., 2007*; *Amsler, McClintock & Baker, 2008*). These densities of amphipods are one to three times higher than those typically reported in tropical and temperate regions (*Amsler, McClintock & Baker, 2014*).

Mesograzer assemblages, particularly amphipods, have a mutualistic relationship with the macroalgae they shelter on (*Amsler, McClintock & Baker, 2014*). Most of the macroalgal community is chemically defended against herbivory, including all of the large browns and many of the common reds with the notable exception of *Palmaria decipiens* (*Amsler et al., 2005*; *Aumack et al., 2010*; *Amsler, McClintock & Baker, 2020*). Instead of consuming
their macroalgal hosts, amphipods consume epiphytic algae and emergent filamentous endophytes from their hosts (*Aumack et al., 2011a*) and gain refuge from predatory fish in return (*Zamzow et al., 2010*). These amphipods are so effective at removing epiphytes that macroalgae in the WAP usually lack visible epiphytic filaments in areas with amphipod grazers (*Peters, 2003*; *Amsler et al., 2009*). The removal of filamentous algae improves the overall health of the macroalga by removing competition for light that arises when the filaments grow out of the algal thallus (*Aumack et al., 2011a*).

A significant shift in the amphipod-macroalgal dynamic could have major consequences on the structure of the benthic community (*Rodriguez & Saravia, 2024*). Changes to the environment, such as OA, could impact the relationship between macroalgae and amphipods. Crustaceans located in polar regions are likely sensitive to OA due to these species having historically low metabolic activity and low temperature habitats (*Clarke, 1998*; *Whiteley, 2011*). Amphipod communities in polar regions have shown assemblage reorganization in lowered pH waters by increasing the relative abundances of copepods and ostracods and decreasing the relative abundance of the amphipods *Metaleptamphopus pectinatus* and *Oraderea* spp. (*Schram et al., 2016a*). Part of this reorganization may be occurring due to physiological challenges faced by amphipods. A similar OA study by *Schram et al. (2016b)* found the most noticeable spikes in mortality among the amphipods *Gondogeneia antarctica* and *Paradexamine fissicauda* maintained under OA conditions corresponded with the time of highest molt frequency.

More information is needed to understand why some amphipod species in these assemblages are more resistant to OA than others. The assemblage reorganization observed in *Schram et al. (2016a)* occurred under a short term OA exposure of 30 days. Additionally, another experiment by *Schram et al. (2016b)* found that the impacts of OA on two Antarctic amphipods became more pronounced under a longer exposure time of 90 days. Furthermore, the effects of OA were relatively stronger on one species, *Paradexamine fissicauda*. It is currently unclear if the reorganization observed in *Schram et al. (2016a)* would be maintained or strengthened under longer term exposure. In the present study, we hypothesized that amphipod species from the WAP would vary in their survival under longer-term OA conditions. We assessed the survival of different members of a macroalgal-associated amphipod assemblage by maintaining the assemblages under three different pH treatments simulating ambient conditions (approximately pH 8.1), near-future conditions for 2100 (pH 7.7; *IPCC, 2022*), and distant future conditions (pH 7.3; *IPCC, 2022*) for 52 days then enumerating the species. This assessment was used to determine which amphipod species were comparatively more resistant to OA.

## MATERIALS & METHODS

### Collection of macroalgae and mesograzer assemblages

The brown macroalga *Desmarestia menziesii* J. Agardh and its associated mesograzer assemblage were collected from five sites near the US Antarctic Research Program's Palmer Station on Anvers Island on the western Antarctic Peninsula. *Desmarestia menziesii* was collected by scuba from Palmer Station pier (S64°46.477′, W64°03.274′), Amsler Island

(S64°45.629′, W64°05.879), Litchfield Island (S64°46.095′, W64°03.025′), Stepping Stones (S64°47.031′, W63°59.453′), and Christine Island (S64°47.479′, W64°01.024′) between 4 and 11 m on 21–23 January 2020. Collections were made from multiple sites so that the amphipod assemblages from them were more generally representative of those in the overall Palmer Station area. *Palmaria decipiens* (Reinsch) R. W. Ricker, which was used as a supplementary food source only, was collected on small rocks from the Palmer Station pier on 20 and 21 January 2020 and from Hero Inlet (S64°46.555′, W64°02.944′) on 23 January 2020.

Amphipods for the experiment were collected using the methods described by *Huang et al. (2007)*. Briefly, *D. menziesii* thalli were cut with a knife then gently floated into a fine mesh bag to avoid disturbing the associated grazers. Collection bags were immediately transported to Palmer Station in 19-L buckets filled with seawater. Grazers were removed from *D. menziesii* by repeatedly rinsing the alga with seawater in a series of fine mesh bags submerged in a tank of seawater. After the final rinse, any remaining grazers were removed by hand.

The collected fauna from *D. menziesii* was pooled to create a sample representative of the local grazer assemblages around Palmer Station. Individuals of the carnivorous amphipod *Bovallia gigantea* Pfeffer, 1888 larger than approximately 1.5 cm were removed from the pool to prevent predation on other species during the experiment. The sample was divided into thirty-two equal aliquots with a Folsom plankton splitter to generate an approximate grazer number and diversity representative of that associated with 85 g of *D. menziesii* in the original collections. Aliquots were randomly assigned and placed in either one of the twenty-four experimental buckets or one of the eight initial samples. All of the initial samples were immediately preserved and species were enumerated.

## Experimental setup

Two adjacent aquarium tanks (2.5 m diameter and 1 m depth; 3800 L) in the aquarium facility at Palmer Station were equipped with twenty-four 19-L white plastic buckets (see Fig. S1 and *Oswalt et al., 2025*). The experimental setup was maintained with 24-h constant light consistent with light availability at the time of collection. An ambient flow-through seawater bath was plumbed for both tanks to maintain an ambient temperature in all experimental buckets. Nylon mesh (75 μm) was fixed to the outflow pipes in the buckets to prevent grazers from flowing out of the experiment.

An aliquot of the grazer assemblage and an approximately 85 g wet weight of *D. menziesii* thallus was placed in each bucket. To provide a food supply for the grazers, *D. menziesii* were seeded with diatoms and held in outdoor tanks with unfiltered seawater for over three weeks to allow the growth of an epiphytic diatom assemblage on the thalli. Plankton mesh (63 μm, regularly cleaned) was placed over the water inlets and standpipe drains to prevent additional mesograzers entering buckets with the unfiltered water. The epiphytes were grown in higher abundance than observed *in situ* to ensure a consistent food supply over the experiment. In addition to *D. menziesii*, a *P. decipiens* blade was placed into each bucket as an alternative food source for the assemblage as some, but not all, of the

amphipod species consume this non-chemically defended red alga (*Amsler, McClintock & Baker, 2020*).

A common head tank dispersed filtered seawater into a water distributor located in the center of each aquarium tank. Water flowed into twelve mixing reservoirs from a distributor located in the center of each aquarium tank. In addition to the water inflow pipe, each mixing reservoir also contained a pH probe (model 59001-70; Cole Parmer), a tube with mixed air and compressed $CO_2$, and a tube with compressed air alone. Seawater pH was monitored and adjusted to the desired pH in the mixing reservoirs then flowed into the experimental buckets before flowing out of the buckets and into the seawater bath.

Eights buckets were assigned for each pH treatment (ambient (8.1), 7.7, and 7.3) based on the average pH recorded for Palmer Station during collection (approximately 8.1) and the predicted pH levels for the near and distant future (*IPCC, 2022*). Replicates in the decreased pH treatments were individually regulated with an automated pH monitoring system (*Schoenrock et al., 2016*; *Schram et al., 2016a*). Bucket pH was lowered and maintained by bubbling an air-$CO_2$ mixture or air alone into the mixing reservoirs as needed. A Multi-tube Gas Proportioning rotameter (Omega Engineering, Inc., Stamford, CT, USA) connected to multiple air pumps and a $CO_2$ cylinder combined ratios of air and $CO_2$ to create the air-$CO_2$ mixture. Half of the replicates for each treatment were randomly assigned to each water bath to control for differences in light availability due to the location of the water baths. Gas exchange with the atmosphere was reduced by fixing a clear Plexiglass cover to each bucket (*Schram et al., 2016a*).

Each day, one quarter of the buckets had seawater samples collected for pH (determined spectrophotometrically) and seawater total alkalinity (TA) measurements (determined by potentiometric titration; *Dickson, Sabine & Christian, 2007*). This resulted in all buckets being tested over a four-day cycle. Additionally, the pH of every bucket was monitored daily with a hand-held pH probe (Orion Star A221 pH meter; Thermo Fisher Scientific, Waltham, MA, USA).

After the grazers and macroalgae were placed in the buckets, the pH was decreased in the lower pH treatments over a 28-hour period. Following a 52-day exposure, *D. menziesii* were removed from the buckets. Each alga was successively rinsed in 19-L buckets of seawater to remove the grazers and subjected to a visual inspection to ensure grazers were fully removed. The collected mesograzer assemblages were preserved in 10% formalin and later identified to the lowest taxon possible using a dissecting microscope and enumerated.

## Carbonate chemistry determination

Seawater pH was determined using a Perkin Elmer UV/VIS Spectrometer Lambda 40P on the total hydrogen scale ($pH_T$) after the addition of m-cresol purple, a pH sensitive indicator (*Dickson, Sabine & Christian, 2007*). TA was measured using open cell potentiometric titration (SOP 3b, *Dickson, Sabine & Christian, 2007*). Seawater samples were maintained at 20 °C using a Neslab RTE-7 Circulating Bath (Thermo Fisher Scientific). Titrations were completed utilizing a Mettler-Toldeo T50 open cell titrator equipped with a pH probe (Model DGi 115-SC) while Mettler-Toledo LabX® software recorded titrant volumes in real time. Temperature, salinity, pH, and TA data were used to calculate carbonate

chemistry parameters. CO2calc software (*Robbins et al., 2010*) with $CO_2$ constants from *Roy et al. (1993)* and a $KHSO_4$ acidity constant from *Dickson (1990)* were used for carbonate calculations. Salinity was measured with a Seabird 45 MicroTSG from the Palmer Station Waterwall (*Palmer Station Instrument Technician, 2023*) and temperature was recorded during seawater sample collection.

## Statistical analyses

Univariate and multivariate statistical analyses were performed with R v4.4.0 (*R Core Team, 2024*) using the *vegan* (*Oksanen et al., 2024*) and *stats* package (*R Core Team, 2024*). Normality and homogeneity of variance were tested using the Shapiro–Wilk test and Levene's test, respectively. A one-way analysis of similarity (ANOSIM) quantified similarities of the amphipod assemblages across the pH treatments. A one-way analysis of variance (ANOVA) was performed to examine the total abundance of the mesograzer assemblages between the experimental groups. A permutational analysis of variance (PERMANOVA) was used to examine the composition of taxa between the experimental groups. These data were arcsine transformed to help normalize percentages. The cutoff for statistical significance was set to $\alpha \leq 0.05$.

Nonparametric, multivariate analyses using PRIMER-e v. 7 (Quest Research Limited) were performed to compare macroalgal and overall species assemblages across sample sites with statistical methodology following recommendations of *Clarke et al. (2014)*. Because species numbers varied by several orders of magnitude across samples, these data were square-root transformed to down-weight the influence of the most abundant species. To visually compare similarity among initial and experimental amphipod assemblages, non-metric multidimensional scaling (nMDS) ordination plots were created based on Bray–Curtis similarity matrices. Statistical differences were determined using CLUSTER analysis with similarity profile (SIMPROF) tests ($\alpha = 0.05$). To further visually compare amphipod assemblages in the pH treatments, bootstrap averages and bootstrap regions were calculated within a metric multidimensional scaling (mMDS) ordination plot created on Bray–Curtis similarity matrices.

## RESULTS

Seawater parameters during the experiment are summarized in Table 1. Mean ($\pm$ SD) pH values of the lowered pH treatments were close to their target pH. Data for each parameter (pH, total alkalinity, salinity, DIC, *etc.*) on each day of the experiment are available at the US Antarctic Program Data Center (see Details of Data Deposit statement, below). Total alkalinity, temperature, and salinity remained similar between the pH treatments.

No significant differences between the amphipod assemblages at different pH was revealed by an ANOSIM. This type of test produces an R-statistic that ranges from $-1$ and 1 with values close to 0 indicating no difference between the groups. The data in the present study produced an R-statistic of $-0.02$, demonstrating that assemblages between the pH treatments were very similar to each other. The nMDS analyses also did not show a significant difference in the pH treated assemblages but did show a significant difference (SIMPROF test) between the initial samples and treatment groups (Fig. 1). However, the

**Table 1  Carbonate chemistry of the ambient and lowered pH treatments (mean ± SD).** Seawater parameters ($n = 8$) calculated from TA ($\mu$mol kg$^{-1}$ SW), spectrophotometric pH$_T$ (mean ± SD), temperature (° C), and salinity (ppt). Calculated parameters included $p\,CO_2$ ($\mu$atm) and saturation states of aragonite ($\Omega_{arg}$) and calcite ($\Omega_{cal}$).

|  | Ambient | pH 7.7 | pH 7.3 |
|---|---|---|---|
| pH$_T$ | 8.07 ± 0.07 | 7.69 ± 0.11 | 7.32 ± 0.08 |
| TA | 2,284 ± 78 | 2,261 ± 42 | 2,245 ± 64 |
| Temp | 2.45 ± 0.34 | 2.46 ± 0.34 | 2.46 ± 0.33 |
| Salinity | 32.97 ± 0.29 | 32.96 ± 0.29 | 32.96 ± 0.29 |
| $p\,CO_2$ | 367 ± 48 | 947 ± 221 | 2,245 ± 409 |
| DIC | 2,145 ± 56 | 2,243 ± 41 | 2,343 ± 72 |
| $\Omega_{arg}$ | 1.62 ± 0.37 | 0.73 ± 0.21 | 0.32 ± 0.07 |
| $\Omega_{cal}$ | 2.59 ± 0.58 | 1.17 ± 0.33 | 0.50 ± 0.11 |

bootstrap analysis of the mMDS data (Fig. 2, Video S1) did reveal that the amphipod assemblages were beginning to separate into distinct pH groups when analyzed at the completion of the 52-day experiment. The total abundance of all mesograzers and total abundance of amphipods were found to be normally distributed, indicating parametric tests could be utilized. The abundance of each taxon was not normally distributed. As a result, the non-parametric PERMANOVA was utilized to analyze these data. The total abundance of all mesograzers and total abundance of amphipods both generally decreased with decreased pH but neither trend was significant (Fig. 3, $F_{2,21} = 0.549$, $p = 0.59$, $F_{2,21} = 2.241$, $p = 0.13$).

The proportions of amphipods in the initial assemblages (Fig. 4, $p < 0.05$ SIMPROF test) were significantly different compared to all of the treatment groups. The relative proportions of each taxon across the pH treatments were not found to be significantly different (Fig. 5, PERMANOVA: $F_{2,21} = 0.825$, $p = 0.62$). However, some general, nonsignificant trends were observed. *Djerboa furcipes* Chevreux, 1906 and *Bovallia gigantea* both showed resilience to decreased pH. *Bovallia gigantea* had a similar abundance across the pH treatments while *D. furcipes* generally increased in proportion with decreasing pH. *Gondogeneia antarctica* Chevreux, 1906 and *Metaleptamphopus pectinatus* Chevreux, 1912 both fell into an intermediate group. *Gondogeneia antarctica* experienced a decrease in its relative abundance from ambient pH to pH 7.7. However, its relative abundance was similar between the pH 7.7 and pH 7.3 treatments. *Metaleptamphopus pectinatus*, in comparison, initially increased in relative abundance as pH decreased to 7.7, but its relative abundance decreased to a similar level as in the ambient treatment at pH 7.3. Finally, both *Oradarea* spp. and *Prostebbingia gracilis* Chevreux, 1912 slightly decreased in abundance with decreased pH.

## DISCUSSION

Macroalgal-associated mesograzer assemblages were exposed to three pH treatments simulating current ambient conditions (pH 8.1), end of the century conditions (pH 7.7, IPCC, 2022), and distant future conditions (pH 7.3, IPCC, 2022) for 52 days. There was a

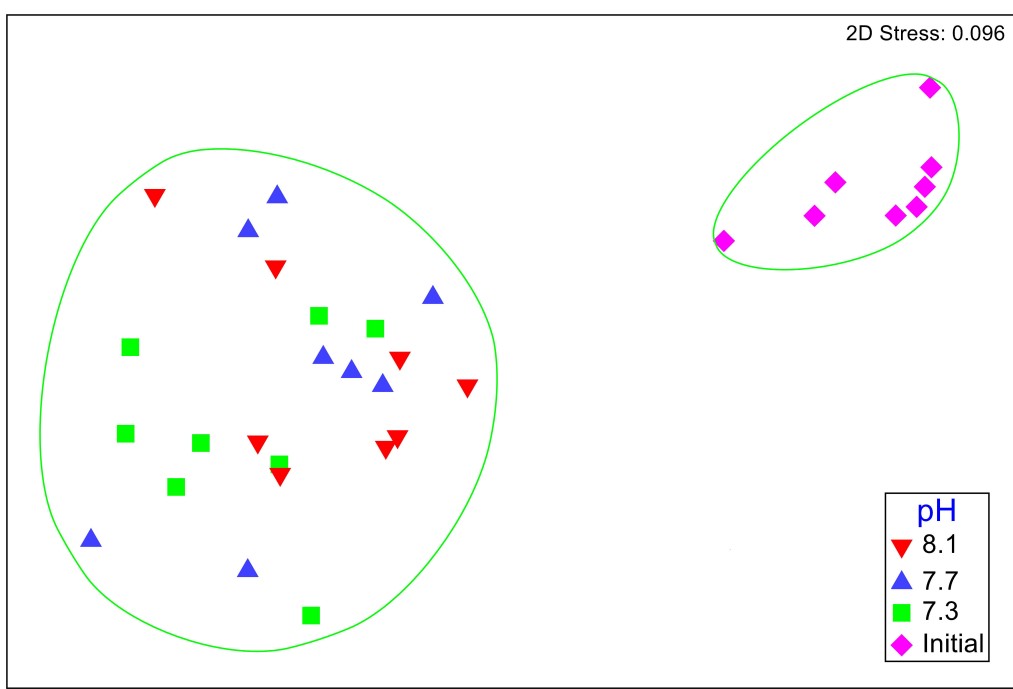

**Figure 1** **Nonmetric multidimensional scaling (nMDS) plot based on Bray–Curtis similarity matrix for mesograzer assemblages without copepods or ostracods maintained at pH 8.1, pH 7.7, and pH 7.3 for a 52-day exposure period and initial samples.** The green circles represent groups that are not significantly different ($p > 0.05$) in a SIMPROF test following CLUSTER analysis.

significant difference between the initial assemblages and the treatment assemblages. This difference was likely due to the large amount of mortality of the amphipod *M. pectinatus* during the experiment. In the initial assemblages, *M. pectinatus* constituted approximately 65% of the initial amphipod assemblage (Fig. 4). However, *M. pectinatus* constituted between 11–19% of the treatment assemblages. There was also a large amount of variation observed in the initial samples (see Table S1 for species counts in each initial sample). For example, the number of *M. pectinatus* ranged between 176–388 individuals and *Oradarea* spp. counts ranged from 25–80 individuals depending on the initial sample. While efforts were made to make the initial assemblages as similar as possible, these variations in the assemblages could have had an impact on the experiment's results.

Although all experimental treatments were significantly different from the initial assemblages, there was no significant difference in total abundance or species proportion between any of the pH treatments. However, since the amphipod assemblages among the treatments had become somewhat divergent in a bootstrap analysis, it is possible that they could have become significantly different under a longer exposure time if the experiment had not been prematurely ended due to the COVID-19 pandemic. Some trends in relative amphipod abundances were noted, but many of these trends were not consistent with the findings of a similar study by *Schram et al. (2016a)*. Both our study and *Schram et al. (2016a)* found that the abundance of *B. gigantea* stayed consistent across the pH treatments.

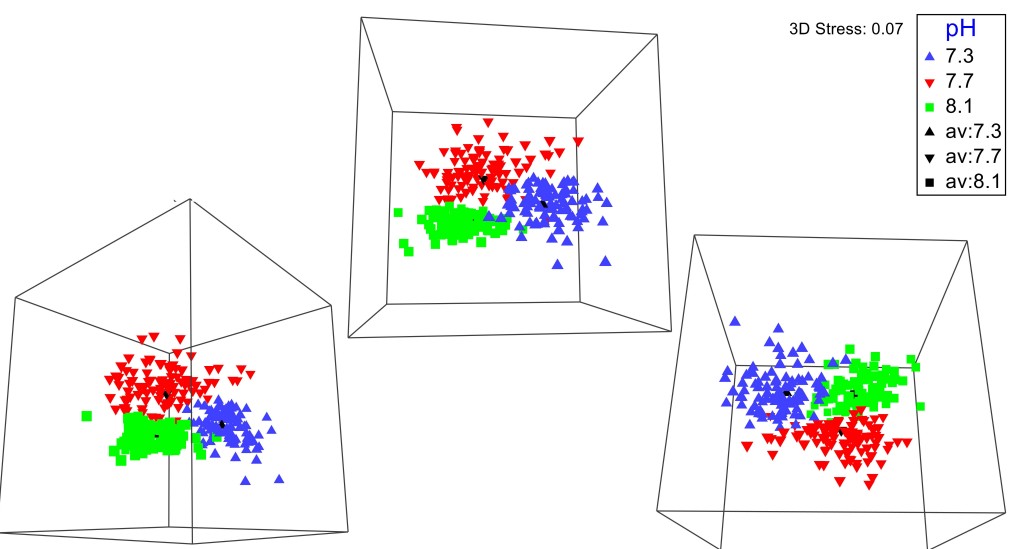

**Figure 2  Three different 3-D perspectives of a metric multidimensional scaling (mMDS) bootstrap analysis for amphipod assemblages maintained at pH 8.1, pH 7.7, and pH 7.3 for a 52-day exposure period.** The black markers indicate average values for each pH treatment. The colored markers do not represent data points but rather bootstrap regions, analogous to error bars on plots of univariate data. See also Video S1 of the plot being rotated in space.

A decrease in the abundance of *Oradarea* spp. between the ambient and pH 7.3 treatment was also found in both studies. The proportion of *M. pectinatus* in this study increased in the pH 7.7 treatment but was similar between the ambient and pH 7.3 treatments. *Schram et al. (2016a)*, however, found a significant decrease of *M. pectinatus* with decreased pH. Opposite trends were found in *P. gracilis* with the species decreasing with decreased pH in our study but increasing with decreased pH in *Schram et al. (2016a)*.

One possible explanation for the discrepancy between the two studies is the starting assemblage organization. Although both studies sampled in similar locations using the same methods, species compositions varied. For example, *D. furcipes* constituted a large proportion of our assemblages, approximately 11% in the ambient treatment, but was not present in *Schram et al. (2016a)*. Furthermore, *M. pectinatus* and *Oradarea* spp. abundances were found to be ten-fold and two-fold higher, respectively, in the assemblages of *Schram et al. (2016a)* compared to this study. The starting assemblage composition could have had an impact on the final assemblage composition in the experimental treatments. Collection times varied between the two studies with collections occurring in January 2020 in the present study and in March 2013 in *Schram et al. (2016a)*. Seasonal or interannual differences in species abundances could explain why the starting assemblage composition varied between the two studies.

Another possible explanation for these varied results could be the length of the experiment. The experiment in *Schram et al. (2016a)* ran for 30 days while our experiment ran for 52 days. This difference in experiment time could explain why some species, like *M. pectinatus*, were so comparatively low in our experiment. Our initial samples contained,

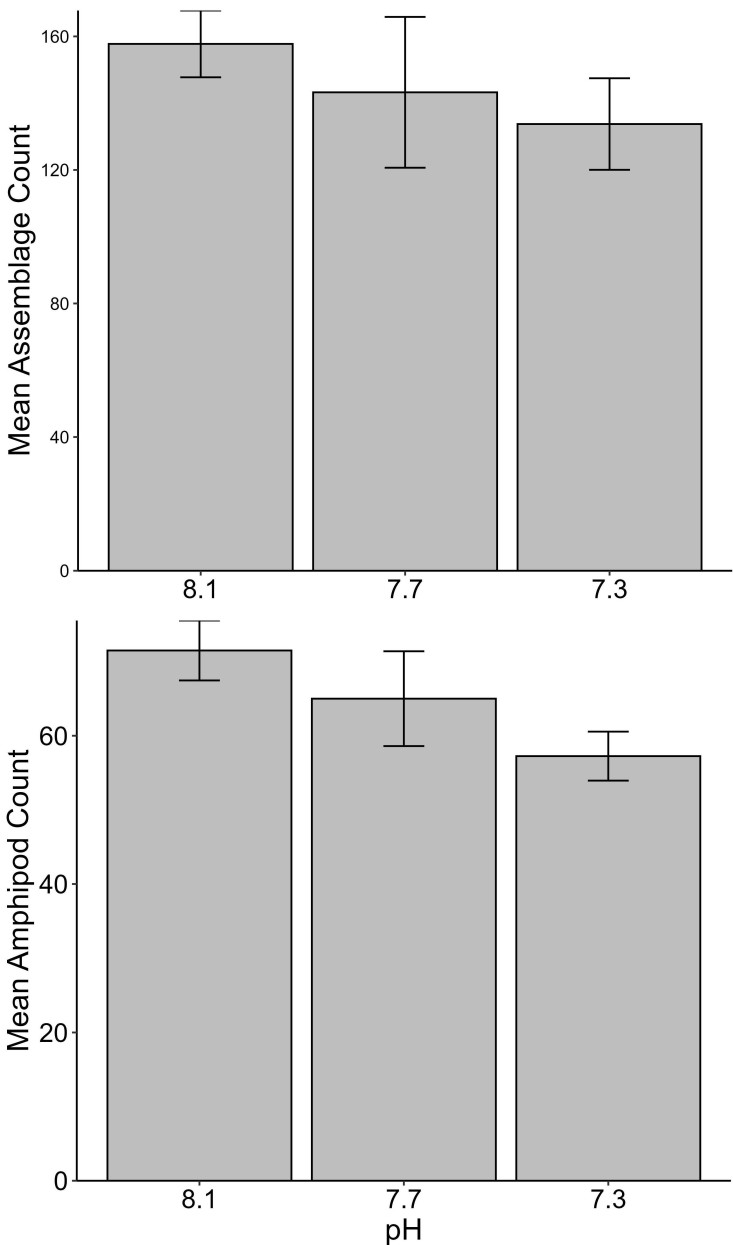

**Figure 3** Total assemblage and amphipod counts (mean ± SE) maintained at pH 8.1, pH 7.7, and pH 7.3 for a 52-day exposure period ($n = 8$).

on average, over 300 *M. pectinatus*. However, most of the final assemblages across each of the experimental treatments contained less than fifteen *M. pectinatus*, demonstrating a massive amount of mortality from being held in the experiment. Part of this mortality could be due to the size of *M. pectinatus*. This species is generally small in size, making it vulnerable to predation by the small number of *B. gigantea* in the experiment. Final assemblage composition could have been impacted by increased mortality from being held

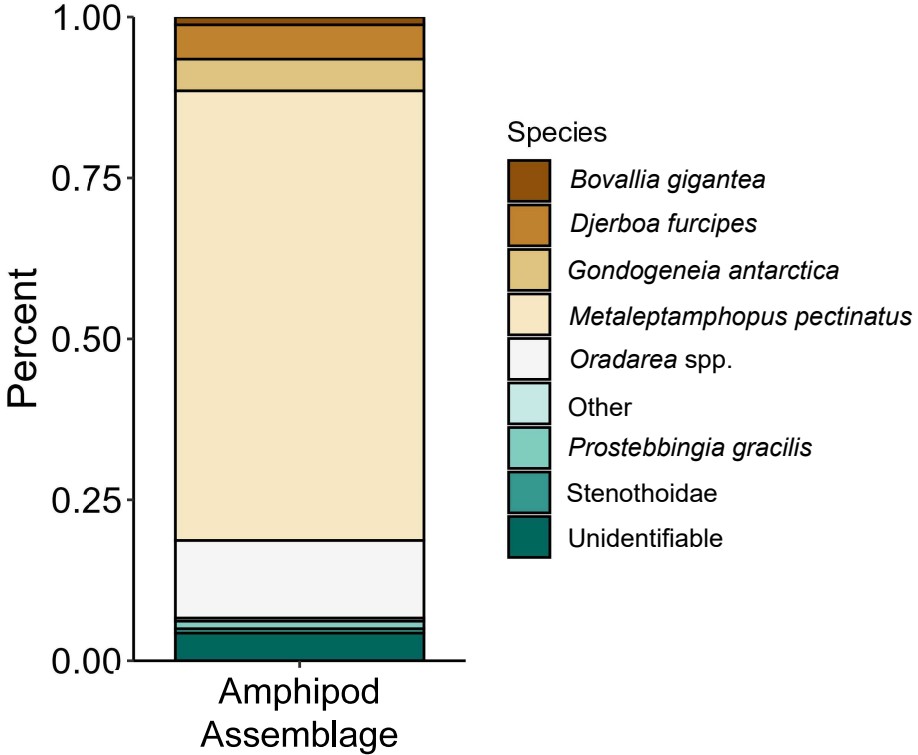

**Figure 4** **Relative abundance of amphipods from the initial assemblages ($n = 8$).** The group labeled 'other' consisted of species that constituted less than 1% of the assemblage. This group included *Jassa* sp., *Prothaumatelson nasutum*, *Paraphimedia integricauda*, *Gnathiphimedia* sp., Lysianasidae, and *Probolisca ovata*.

in the experiment for a longer period of time. This reduction in *M. pectinatus* could also partially explain why our results differed from *Schram et al. (2016a)* since a majority of the dissimilarity of the pH 7.3 assemblage in their experiment was driven by low *M. pectinatus* abundance.

Our results indicate that amphipod assemblages associated with *D. menziesii* exhibit resistance to long-term exposure to near future and distant future OA conditions. These results are in contrast to the reduction in species richness and abundance of crustacean assemblages with decreased pH (*Hale et al., 2011*; *Kroeker et al., 2011*; *Fabricius et al., 2014*). However, mesocosm experiments have been found to be less sensitive in detecting species replacements, community reshuffling, or biodiversity changes in response to OA compared to natural systems (*Nagelkerken & Connell, 2022*). Our results indicate that WAP amphipod communities could be protected by the insurance effect by the preservation of biodiversity within their assemblages (*Yachi & Loreau, 1999*; *Rastelli et al., 2020*).

A longer experiment is likely necessary to gain a better understanding of how OA impacts macroalgal-associated amphipod assemblages. Originally, we planned on having a longer exposure period for the amphipod assemblages. The unexpected COVID-19 pandemic forced us to end the experiment prematurely. Even with this shortened experiment time,

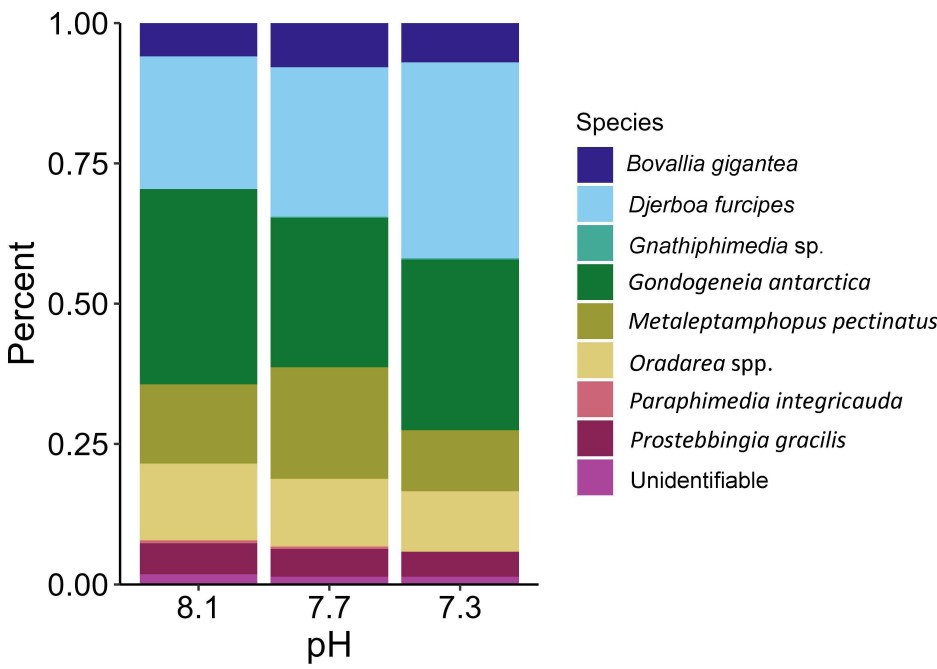

**Figure 5** Relative abundance of amphipods maintained at pH 8.1, pH 7.7, and pH 7.3 for a 52-day exposure period ($n = 8$).

bootstrap nMDS data (Fig. 2) show that the assemblages were beginning to separate into separate groups, even if this result was not significant. A significant increase in mortality of the amphipods *G. antarctica* and *P. fissicauda* has been found after a three-month exposure to OA conditions (*Schram et al., 2016b*). The mortality of *G. antarctica* in this longer experiment was higher than the mortality reported in *Schram et al. (2016a)*, demonstrating that exposure time has an impact of the severity of OA affects for these assemblages. Furthermore, peaks in mortality in *Schram et al. (2016b)* coincided with peaks in molt frequency which could indicate that OA impacts on other physiological processes, like molting, likely impact overall survival.

While the results of this experiment show that amphipod assemblages were not impacted in relative mortality, this does not mean that the assemblages were completely unaffected to OA conditions. For example, adult Antarctic krill can maintain their survival, growth, and respiration rate under 2,000 $\mu$atm $pCO_2$ exposure (*Ericson et al., 2018*). However, hatch rates and embryo survival of Antarctic krill decreases by over 90% under the same $CO_2$ conditions (*Kawaguchi et al., 2011*), demonstrating there may be unforeseen long-term impacts on species that are identified as more resistant to OA in shorter studies. Hypothetically, amphipod assemblages could also become more susceptible to other environmental changes, such as ocean warming. Although the current experiment only controlled pH and let temperature and salinity change naturally, these two factors did vary over the course of the experiment. It is conceivable that stress from decreased pH could leave the amphipods more sensitive to temperature variations. For example, warming can have

an additive effect on the consumption rate of the amphipod *P. fissicauda* with decreased pH and a nonsignificant additive increase on the mortality of *P. fissicauda* and *G. antarctica* (*Schram et al., 2016b*). The temperature difference in *Schram et al. (2016b)* was larger than the variability in the present experiment, but there was variability throughout the experiment.

In some cases, the severity of OA effects is reliant on the amount of time an organism has to acclimate. The sea urchin *Strongylocentrotus droebachiensis* experiences a 4.5-fold decrease in fecundity and a 5–9-fold decrease in offspring reaching the juvenile stage when exposed to four months of decreased pH. However, there was no difference in either fecundity or offspring survival when the sea urchins are exposed to decreased pH for sixteen months compared to ambient conditions (*Dupont et al., 2013*). In addition to longer exposure across one individual's life, transgenerational exposure can have positive and negative effects on a species. *Lopes et al. (2019)* found that exposure to OA conditions decreased the amphipod *Gammarus locusta* parental generation's survivability and caused DNA damage in their offspring. Furthermore, the offspring that could survive OA were incapable of returning to ambient conditions without experiencing an increase in lipid damage and death. In some cases, transgenerational exposure increases an organism's ability to withstand decreased pH. Some invertebrates can experience better extracellular pH regulation, have larger eggs, and have more resilient offspring when exposed to OA conditions for multiple generations (*Parker et al., 2015*; *Zhao et al., 2018*; *Zhao et al., 2019*). Most Antarctic amphipods are relatively long lived (*Bone, 1972*; *Thurston, 1972*; *Brown, King & Harrison, 2015*), making any differences observed in the present study more likely due to differences in survival rather than reproduction. Although 52 days can be a large portion of the lifespan of many lower latitude amphipods, this is not true for Antarctic species, which commonly live for several years (*Bone, 1972*; *Thurston, 1972*; *Bluhm, Brey & Klages, 2001*; *Brown, King & Harrison, 2015*). Because of this, the results from the present study cannot illuminate how OA will impact amphipod assemblages on generational time scales.

Mesograzer assemblages along the WAP may be preconditioned to tolerate decreases in pH. Seawater pH along the WAP can fluctuate up to 0.6 pH units annually, ranging from approximately 8.6 in December to 8.0 in May (*Schram et al., 2015*). There is a growing theoretical framework for how organisms will respond to climate change based on the magnitude and predictability of environmental fluctuations. Phenotypic plasticity and bet-hedging are the two most common adaptations that arise from fluctuating selection, but the type of adaptation that evolves is dependent on the timescale of fluctuations (*Tufto, 2015*). Frequent and predictable environmental change supports the development of plastic adaptations (*Botero et al., 2015*). This concept has also been seen experimentally in mites and mussels to predictable fluctuations in temperature and pH, respectively (*Deere et al., 2006*; *Bitter et al., 2021*). The amphipod assemblages examined in the present study may have high tolerances to pH fluctuations because they are found in environments that are known to have large pH fluctuations throughout the year (*Schram et al., 2015*).

Amphipods may be benefiting from their close relationship with macroalgae. Seaweeds have boundary layers that can range from 0.1 to 10.2 mm thick depending on the species

and surrounding water flow (*Raven & Hurd, 2012*). These boundary layers can serve as a refuge for calcifying species during the day by buffering seawater pH (*Hurd et al., 2011*). The pH within the boundary layers is controlled by seaweed metabolism. During the day, pH tends to increase in the boundary layer as photosynthesis occurs. At night, pH decreases as algae continue to undergo respiration (*Hurd, 2015*). The pH within the boundary layers of macroalgae and seagrasses can be 0.07 to 1.2 pH units higher than surrounding seawater during the day (*Jones, Eaton & Hardwick, 2000*; *Krause-Jensen et al., 2015*; *Hendriks et al., 2017*). Experimentally, macroalgae have been found to mitigate some negative effects of OA on associated calcifiers. The addition of the green alga *Ulva* in high $CO_2$ treatments was found to increase saturation states of aragonite and calcite and increased the growth rates of clams, scallops, and oysters (*Young & Gobler, 2018*). *Wahl et al. (2018)* found that the brown alga *Fucus vesiculosus* can act as a temporal refuge from OA conditions to the mussel *Mytilus edulis*. The mussels were able to maintain high calcification in low pH treatments by shifting a majority of their calcification process to the daytime when algal photosynthesis was occurring, and pH and calcite saturation were higher. In the present experiment, *D. menziesii* could have been acting as a refuge for the amphipods. Live *D. menziesii* thalli were maintained with 24-hour constant light consistent with the time of collection throughout the experiment. Photosynthesis should have been occurring continuously throughout the entire exposure period. The amphipod assemblages could have been benefiting from a possible increase in pH in the alga's boundary layer, possibly explaining why no significant difference in mortality was found for the total assemblage or within a species between the different pH treatments.

## CONCLUSIONS

The results of the present study show that invertebrate mortality of a macroalgal-associated assemblage is not negatively impacted by OA. The assemblages between the pH treatments were similar in total assemblage number and assemblage composition. These results differ from previous studies (*Schram et al., 2016a*) and demonstrate that starting assemblage composition or exposure time could impact assemblage resistance to OA. Furthermore, the close association with these assemblages to *D. menziesii* could be mitigating some of the direct negative impacts of OA. Overall, our results suggest that *D. menziesii*-associated amphipod assemblages may be resistant to long-term OA exposure.

## ACKNOWLEDGEMENTS

The authors gratefully acknowledge the science and logistical support staff of Antarctic Support Contract for their help and support of the United States Antarctic Program. The manuscript was improved based on comments from Jeff Morris, Stacy Krueger-Hadfield, Farzaneh Momtazi, and two anonymous reviewers.

### Funding
This research was supported by National Science Foundation award OPP-184887 from the Antarctic Organisms and Ecosystems Program. The funders had no role in study design, data collection and analysis, decision to publish, or preparation of the manuscript.

### Grant Disclosures
The following grant information was disclosed by the authors:
National Science Foundation Antarctic Organisms and Ecosystems Program: OPP-184887.

### Competing Interests
The authors declare there are no competing interests.

### Author Contributions
- Hannah E. Oswalt conceived and designed the experiments, performed the experiments, analyzed the data, prepared figures and/or tables, authored or reviewed drafts of the article, and approved the final draft.
- Julie B. Schram conceived and designed the experiments, performed the experiments, authored or reviewed drafts of the article, and approved the final draft.
- Margaret O. Amsler conceived and designed the experiments, performed the experiments, authored or reviewed drafts of the article, and approved the final draft.
- Charles D. Amsler conceived and designed the experiments, performed the experiments, analyzed the data, prepared figures and/or tables, authored or reviewed drafts of the article, and approved the final draft.
- James B. McClintock conceived and designed the experiments, performed the experiments, authored or reviewed drafts of the article, and approved the final draft.

### Data Availability
The data are available at the United States Antarctic Program Data Center: https://www.usap-dc.org/view/project/p0010193.

### Supplemental Information
Supplemental information for this article can be found online at http://dx.doi.org/10.7717/peerj.19368#supplemental-information.

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
