# Peer review of "Antarctic macroalgal-associated amphipod assemblages exhibit long-term resistance to ocean acidification"

_PeerJ, doi:10.7717/peerj.19368_

## Round 0.1 · original submission · Major Revisions

The three reviewers all liked your MS and made thoughtful and constructive suggestions on it, which included clarifying the aims and contribution of the study in the Introduction and Discussion, and providing extra details in the Methods. Please address these and the reviewers' other comments in your revision.

General comments from Reviewer 3:

"The manuscript we reviewed explores how 52-day low pH exposures affect amphipod assemblages with macroalgae in the Western Antarctic Peninsula (WAP). The manuscript is well-written, using professional yet clear language. The methodology, including seawater system maintenance and experimental designs, is described in a comprehensible manner. The figures in the results section are well-supported with clear statistical descriptions. However, some parts should be edited to enhance the paper’s conciseness and improve the experiment’s replicability to fully meet PeerJ’s expectations.

The authors thoroughly discuss the significance of low pH for calcified organisms, dedicating a substantial portion of the manuscript to this topic. While the impacts of acidification on calcified organisms are undoubtedly important, overemphasizing this discussion could potentially confuse some readers, as amphipods are not heavily calcified and may exhibit different mechanisms that warrant greater focus. Addressing calcified organisms more concisely could better highlight the specific impacts on amphipods’ physiology, ecology, and interactions with macroalgae, which are already well-addressed in the manuscript, ensuring the paper remains focused on its primary context: the effects of ocean acidification (OA) on amphipods.

Additionally, more data and information on the initial amphipod assemblage samples should be provided, either in the Results section alongside Figures 3 and 4 or as supplementary figures. Initial sample groups show a remarkable difference compared to other treatment groups, as seen in Figure 1, yet the manuscript does not sufficiently explain the causes of these differences. The authors also do not clarify when and how the amphipod assemblages were investigated in the method section, which may reduce the experiment’s replicability.

Overall, we believe this manuscript aligns well with PeerJ’s standards but could benefit from the corrections mentioned above. This study offers valuable insights into the impacts of pH on multiple species and highlights the necessity of studying mutual relationships between grazers and macroalgae. Such relationships play a key role in explaining changes in biodiversity and biomass in the shallow benthic ecosystems of the Southern Ocean. With its clear methodology and potential to inspire future research, this study will significantly contribute to the field by promoting deeper understanding of these systems and encouraging the development of modified methodologies for similar studies."

·

Basic reporting

This article investigates the effect of changes in pH under laboratory conditions on the diversity and abundance of bivalves and macroalgae communities collected from the Antarctic region. The article is in acceptable English and only requires some editing. The overall structure of the article is as expected, the review of articles is well done, and tables and figures are correctly included in the article.

Experimental design

The article is on Journal Scope.
Although the study and experimental design are testable, the most important issue of the article is the lack of a clear research question and the uncertainty of the hypothesis tested during this study. In a previous study (Schram et al.,2016), the effect of pH changes on the crustacean community along with macroalgae in the area was addressed with a same method (just in a shorter period). Although the composition of the community, as mentioned in the discussion section, can be influential in determining the response to pH changes, the authors have failed to clearly explain in different sections the necessity and purpose of conducting the study and describing clearly the project question and objective.
Therefor, in comparison with previous study, the novelty of the project is questionable.

Validity of the findings

The experimental set up are acceptable and statistical analyses could be improved by adding some multivariant analyses such as Principal Component Analysis or MANOVA to define which environmental variables are more correlated with number of specimens (it depends on which question is described for the study). Additionally, the result of mMDS show a clear separation between each pH treatments but not statistically significant. It must be considered for the final decision.

Additional comments

There are some technical comments in manuscript that was addressed in PDF file.

Reviewer 2 ·

Basic reporting

This study is significant for advancing our understanding of how calcifying species adapt to anticipated ocean acidification. Such research is particularly crucial in polar regions, where there is a notable gap in knowledge regarding compensatory mechanisms in response to global changes. Overall, I recommend minor revisions to the manuscript, and have listed the minor changes I suggest below.

The manuscript is well-written. Figures, tables and raw data are well organized and available.

The background is clear and references the appropriate literature. However, the introduction section should be expanded to elaborate on the importance of studying the impacts of ocean acidification in Antarctica. Additionally, it should incorporate more recent studies and reviews that discuss its effects on Antarctic organisms and their compensatory mechanisms, using the following literature as references:

Seo H, Cho B, Joo S, Ahn I-Y, Kim T. 2024. Archival records of the Antarctic clam shells from Marian Cove, King George Island suggest a protective mechanism against ocean acidification. Mar Pollut Bull 200:116052.
Figuerola B, Hancock AM, Bax N, Cummings V, Downey R, Griffiths H, Smith J, Stark JS. 2021. A review and meta-analysis of potential impacts of ocean acidification on marine calcifiers from the Southern Ocean. Frontiers in Marine Science 8:584445.
Hancock, A. M., King, C. K., Stark, J. S., McMinn, A., and Davidson, A. T. 2020. Effects of ocean acidification on Antarctic marine organisms: a meta-analysis. Ecol. Evol. 10, 4495–4514.
Conrad, C. J. and Lovenduski, N. S. 2015. Climate-driven variability in the Southern Ocean carbonate system. – J. Clim. 28: 5335–5350.
Hauri, C., Friedrich, T. and Timmermann, A. 2016. Abrupt onset and prolongation of aragonite undersaturation events in the Southern Ocean. – Nat. Clim. Change 6: 172–176.

Experimental design

For the Materials and Methods, I think including a visual representation of the experimental design would provide a clear and concise overview of the experimental setup (lines 165-170). The authors should also clarify the following aspects:

Lines 142-49: Why was P. decipiens collected only at two (and different) sites compared to D. menziesii? Are these sites similar or dominated by one macroalgal species? Please provide clarification and information on the characteristics of the different sites as they could influence on the results of the experiments.

Lines 148-49: Please add depth.

Line 171: Please include details about the proportion of each species present in the aliquot used in the experiment. Was the species composition consistent across all buckets? The authors later note that the initial assemblage composition might influence the assemblage's resistance to OA. If the proportions varied among replicates, this variation could potentially affect the results. It is important they discuss how differences in initial species proportions may have impacted the outcomes and interpretations of their study.

Lines 204-5: A 28-hour period is short so the term “slowly” should be removed from the sentence. Decreasing the pH in such a short time could have adversely affected the grazers in the low pH treatments from the beginning of the experiment. Ideally, should have been acclimated by initially maintaining them at natural pH (8.1) and then gradually reducing the pH (approximately 0.05 units per day) over the course of a week. Please clarify in the main text why this short period was chosen.

Validity of the findings

All findings are valid, and rigorously backed up by statistical analysis. Conclusions are also well stated but the authors should discuss more some of the following results:

Lines 289-91: Why could this species more resilient to low pH than other species such as Oradarea?

Lines 315-16: Did you observe predation?

Line 324: Did the other mentioned studies performed short- or long-term experiments?

Additional comments

Lines 281-82: What does it mean the experiment could be underpowered? Please clarify it.

Line 286: “…if logistical constraints…”?

Reviewer 3 ·

Basic reporting

no comment

Experimental design

no comment

Validity of the findings

no comment

Additional comments

no comment

Annotated reviews are not available for download in order to protect the identity of reviewers who chose to remain anonymous.

---

## Round 0.2 · accepted · Accept

Both reviewers were happy with the revisions you made in response to their comments on the originally submitted manuscript. Congratulations on the article's acceptance!

·

Basic reporting

This article investigates the effect of changes in pH under laboratory conditions on the diversity and abundance of bivalves and macroalgae communities collected from the Antarctic region. The article is in acceptable English and only requires some editing. The overall structure of the article is as expected, the review of articles is well done, and tables and figures are correctly included in the article.

Experimental design

The article is on Journal Scope. The study and experimental design are testable, and I am satisfied with the changes made during the review.

Validity of the findings

The experimental setup is acceptable, and statistical analyses were improved.

Additional comments

I am satisfied with the authors' answers and changes in the article based on the reviewer's comments

Reviewer 2 ·

Basic reporting

The authors did an excellent job incorporating my suggestions in the Introduction.

Experimental design

The authors effectively addressed the methodological issues, which were my primary concern.

Validity of the findings

The revised version is significantly improved compared to the original discussing their results. Given these enhancements, I recommend accepting this important manuscript.